# Ecogeographical Adaptation Revisited: Morphological Variations in the Plateau Brown Frog along an Elevation Gradient on the Qinghai–Tibetan Plateau

**DOI:** 10.3390/biology10111081

**Published:** 2021-10-22

**Authors:** Ka Wah Leung, Shengnan Yang, Xiaoyi Wang, Ke Tang, Junhua Hu

**Affiliations:** 1Chengdu Institute of Biology, Chinese Academy of Sciences, Chengdu 610041, China; liangjh@cib.ac.cn (K.W.L.); yangsn@cib.ac.cn (S.Y.); wangxy1@cib.ac.cn (X.W.); tangke17@mails.ucas.ac.cn (K.T.); 2University of Chinese Academy of Sciences, Beijing 100049, China

**Keywords:** anurans, ecogeographical adaptation, morphology, Qinghai–Tibetan Plateau, *Rana kukunoris*, thermoregulation

## Abstract

**Simple Summary:**

A number of studies have explored how the body size and extremities of frogs vary in response to the changing environmental conditions across different geographical gradients, but the outcomes remain controversial. Here, we studied the morphological variations of the plateau brown frog (*Rana kukunoris*) along an elevation gradient (~1800–3500 m) on the eastern margin of the Qinghai–Tibetan Plateau to understand how environmental and biological factors affect them, and to assess whether these variations help to improve thermoregulation. Although we found that male and female frogs showed different variations in body size and extremities along the elevational gradient, both of them showed a significant decrease in the ratio of extremities to body size with increasing elevation. The decreasing ratio implies a gain of thermoregulatory benefits based on the ecogeographical rules. Moreover, the morphological variations were found to be strongly related to both environmental and biological factors. These results suggest that ecogeographical adaptation in frogs may be more complicated than other terrestrial vertebrate species. Most importantly, the adaptation should be viewed as a result of both environmental and biological factors, while it may also appear as an interactive change between body size and extremities.

**Abstract:**

Several anurans have broad elevational and latitudinal distribution ranges; distinct species and populations may face various environmental and selection stresses. Due to their environmental sensitivity, adaptation is critical for the long-term persistence of anurans. Previous studies have tried to identify the ecogeographical pattern and its mechanism in anurans, suggesting different patterns, but the related explanatory mechanisms are yet to be generally supported and are suggested to be complicated. To explore the elusive mechanisms, we studied the morphological variation of the plateau brown frog (*Rana kukunoris*) along an elevational gradient on the eastern margin of the Qinghai–Tibetan Plateau. Using body size, extremity length, and the ratio between them (extremities/body size) as testing indicators, we examined potential ecogeographical adaptations and investigated how environmental and biological factors could shape the morphological development in *R. kukunoris*. We found that males and females showed different variations in body size and extremities along the elevational gradient, whereas both of them showed a decreasing extremities/body size ratio along elevation. Together with the strong correlations between environmental and biological factors and the morphometrics, we identified ecogeographical adaptation and a sexual difference in the selective pressures on the extremities and body size of the plateau brown frog. Our results imply that geographic variations in anuran morphological traits should be understood as an outcome of environmental and biological factors. Furthermore, ecogeographical adaptation in anurans can manifest as an interactive change between body size and extremities.

## 1. Introduction

Under rapid environmental changes [1,2], the capacity of populations and species to respond adaptively to modified environmental conditions is critical to their long-term persistence [3,4]. Anurans are particularly susceptible to environmental stresses and changes owing to their water-permeable skin [5,6]. The environmental sensitivity from the permeable integument has been a boon and bane to anurans because it could grant an increased rate of gas exchange and osmoregulation on one hand, but a change to the quality of their habitat may interrupt their life cycle on the other [7]. Hence, understanding the relationship between the environment and anurans’ physical or physiological adaptations has raised much attention among ecologists and evolutionary biologists (e.g., [8,9,10,11,12]).

Identifying the pattern and understanding its fundamental process have been a usual practice to deduce the relationship between the environment and species adaptation from the macro view (e.g., [13,14,15,16]). The Bergmann’s rule [17] and the Allen’s rule [18] are two of the most well-known ecogeographical generalizations that stated a trend of changing surface area to volume ratio in endothermic species in response to varied environmental conditions. The Bergmann’s rule and the Allen’s rule, respectively, describe an increasing tendency in body size and a decreasing trend in extremities from warm to cool climates, and from low to high altitude or latitude. While the ecogeographical patterns described by the two rules have been documented in many species (e.g., [15,16]), previous studies failed to reach a consensus on the generalization of these patterns in anurans [8,9,10,19,20,21]. This may infer that the underlying causes and mechanism behind anurans’ ecogeographical adaptation are controversial and likely to be complex. Moreover, studies have shown that anurans could differ markedly in morphology within and between species, due to environmental effects [9,22,23], genetic variations [24,25], and/or sexual dimorphism [26]. Together with the discrepancies occurring at both inter- and intraspecific levels of the ecogeographical patterns in anurans [9,10,22,26], ecogeographical adaptation in anurans may not be a simple consequence of environmental effects. Instead, it may be an outcome between the interaction of environmental adaptation [11,12] and other biological factors [24,26,27].

Numerous studies have shown that the morphological development of anurans is significantly shaped by diverse environmental factors (e.g., [11,12,23]). Anurans’ body growth can be restricted by evapotranspiration, which relates to evaporative water loss through skin [11]. A habitat with abundant resources (primary productivity) would relieve the limitations on the upper limits of body size, by providing enough food sources to support larger anurans [28,29,30]. Moreover, larger anurans survive better in arid environments (low water availability), since their desiccation rate decreases with increasing body mass [11,12]. Despite the extensive evidence for the importance of environmental factors in anurans’ geographic variation [11,12,29,30], many large-scale studies have overlooked the importance of biological factors in anurans’ growth (e.g., [8,9,10]). Moreover, several recent publications have mentioned the joined effect of environmental and biological factors on the ecogeographical adaptations of anurans (e.g., [31,32]). Age and sexual dimorphism are two of the most commonly studied variables that are found to be remarkable factors in affecting anurans’ morphological development [26,33], also when combing with environmental variables [31,32]. As anurans possess indeterminate growth, which allows them to theoretically grow as large as their surrounding environment and diet allow [27], age encompasses an individual’s entire growing period. Moreover, sexual dimorphism represents a result of differentials in growth rate and age at maturity between sexes [34,35]. Therefore, combining both environmental and biological factors in the study of anurans’ ecogeographical adaptation can better aid in our knowledge of the mechanism behind it.

According to the Bergmann’s rule and the Allen’s rule, ecogeographical adaptation is shown in the disproportionate growth of the body and extremities, in correspondence to the varied environment. Body size is the most commonly studied trait in ecogeographical studies of anurans. It is not only affected by climatic conditions [36], but also associated with other important physiological and ecological constraints [37,38]. However, the studies of extremities (e.g., forelimb and hindlimb) in anurans have often been overshadowed by that of body length [25], despite previous studies finding ecological importance in limbs, especially in reproduction [39]. Reproductive benefits in both body size and limbs are seldom reported in the original study targets of Bergmann’s rule and Allen’s rule. This makes environmental stress more direct and dominant in shaping endotherms’ morphological traits compared to anurans, since organisms cannot allocate extra limited resources to reproduction without diverting a proportional amount of energy away from another trait [40]. This might explain why endotherms’ ecogeographical adaptation has been extensively supported [15,41]. However, it has been questioned in anurans, and its process has also been suggested to be more complex [9,10,25]. Due to the reproductive correlation of body size and extremities in anurans [37,42], we argue that ecogeographical adaptation may not simply and independently occur in body size or extremities. There might rather be a joined change between body size and extremities to balance the stress from both thermoregulation and reproduction. To elaborate, a trade-off between reproductive benefits and heat preservation might occur in specific body parts, in which resources are accordingly allocated to the traits for reproductive benefits (e.g., larger body size) and enhanced thermoregulation efficiency (e.g., shorter limbs in cold conditions). In this case, the ratio of extremities to body size (E/BS) might provide information about potential variations in relative selection pressures on body size and extremities resulting from both environmental and biological factors.

In this study, we focused on the morphological variations of the plateau brown frog (*Rana kukunoris*), which is endemic to the Qinghai–Tibetan Plateau (QTP). *R. kukunoris* distributes in Gansu (Gannan, Lanzhou, Qilian Mountains), Ningxia, Eastern Qinghai, Eastern Tibet, and Northwestern Sichuan provinces, occupying a broad elevational range from 700 to 4400 m [43,44]. It has been found to display large variations in morphology and reproduction-related life-history traits [19,21,45]. For example, male *R. kukunoris* prefer large females for mating [43] and female fecundity is positively related to body size [45]. As a terrestrial species, *R. kukunoris* lives and breeds on the edges of various still waters or in humid environments. Their activities are strongly affected by the varied environmental conditions and seasonality on the QTP. Their breeding season ranges from March to May, and, thus, newly metamorphosed frogs are found in June or July. *R. kukunoris* starts hibernation in late September and resumes its active stage in the following March [43]. The extensive elevational gradient, together with the varied environmental conditions across its range, make *R. kukunoris* a model organism for studying ecogeographical adaptation. Previous studies have criticized the importance and necessity of arguing whether the pattern exists, and suggested that ecogeographic study should be focusing on exploring the underlying mechanism rather than simply describing the patterns [41,46]. We therefore proposed and focused on two hypotheses as potential explanations for the enigmatic ecogeographical adaptation in anurans:(1)Ecogeographical adaptation in anurans is evident in an interactive change between body size and extremities (hereafter Hypothesis 1).(2)Geographic variation in anurans’ morphological traits is an outcome of combined environmental and biological effects (hereafter Hypothesis 2).

Following an examination of the presence of sexual dimorphism in *R. kukunori*, we used body size, extremities, and the ratio between them to test for ecogeographical adaptation and Hypothesis 1. We then incorporated environmental (temperature, evapotranspiration, precipitation, and primary productivity) and biological factors (age and sex) to explore how these factors could shape the morphological traits of *R. kukunoris*, as well as to test the validity of Hypothesis 2. Our efforts are to provide a comprehensive view of the mechanisms behind the ecogeographical adaptation of the plateau brown frog. This study provides new insights into anurans’ ecogeographical study and might help to explain the unresolved argument about ecogeographical patterns in anurans.

## 2. Materials and Methods

### 2.1. Data Collection

Field samplings were conducted in the eastern margin of the QTP from late March to early May 2014. We investigated 10 populations of *R. kukunoris* along the elevational gradient from 1797 m a.s.l. (Maoxiang’ping, Maoxian County) to 3453 m a.s.l. (Heihe’qiao, Zoige County; Figure 1; Appendix A).

A total of 255 adult individuals (197 males and 58 females) were measured (Appendix A). We determined the sex of the sexually mature individuals by the absence (female) or presence (male) of nuptial pads on the fore digit [43]. Age determination of each individual was done following the method of Liao and Lu [33] (see also Feng et al. [19]). Morphometric measurements were made by using electronic digital calipers (Kraftwelle company, Hangzhou, China) to the nearest 0.02 mm, which included the following morphological traits defined by Fei et al. [43]: snout–vent length (SVL) represented the body size; lower arm and hand length (LAHL) and hindlimb length (HILL) represented forelimb and hindlimb measurements, respectively (Appendix A). Given the elusiveness of the ecogeographical adaptation, we took the extremity (LAHL + HILL) to body size (SVL) ratio (E/BS) as additional evidence for the existence of ecogeographical adaptation in *R. kukunoris*.

### 2.2. Environmental Variables

Due to the reduced influence of genetic variation among populations under elevational comparison [47], we used the elevation gradient as a geographical reference. We selected four environmental factors since they were found to be influential for morphological variations in anurans [11,12,30,48]. The value of each environmental variable was calculated by the average of the monthly mean based on the individual-specific active period (March to September) over their development years (age). The four environmental variables were (1) monthly mean air temperature (hereafter temperature), which accounted for thigmothermy and heliothermy (heat gain) in amphibians [9,48]; (2) monthly potential evapotranspiration (hereafter PET), which reflected the environmental capacity in water removal owing to both the obtainable evaporative energy and the atmospheric water content [11,49]; (3) monthly normalized difference vegetation index (hereafter NDVI) was used as a proxy of primary productivity and a food availability indicator, predicting greater body sizes of organisms in more productive areas [28,30]; and (4) monthly precipitation (hereafter precipitation) was used as a proxy of water and water–energy balance, representing the limits of water stress to body size [12,50] and to the surrounding vegetation. Since *R. kukunoris* has obvious hibernation activity and a relatively shorter active period compared to anurans living at lower elevations or latitudes [43], we took the species-specific active period over individuals’ development years as the temporal length for the data extraction of environmental variables. It was important to adopt such an approach because taking year-long environmental data might hinder the result of its effect on anurans’ growth, given that their feeding activity and metabolism during hibernation are vastly weakened [51]. We obtained temperature, NDVI, and precipitation at a spatial resolution of 500 × 500 m from the Resource and Environment Science and Data Center (http://www.resdc.cn/, 8 August 2020) and the China Meteorological Data Service Centre (https://data.cma.cn/, 13 August 2020) based on the sampling locations using ArcGIS (v. 10.2.2) and estimated PET by Thornthwaite’s method [52].

### 2.3. Statistical Analysis

Considering that the frequency distribution of the morphological measurements was mostly skewed (except SVL), all measurements were log_10_-transformed to meet the assumption of normality and enhanced homogeneity of variance.

To test for the presence of geographic variation in the environmental condition, we performed linear regression to evaluate the relationships between environmental variables varied and elevation. Previous research showed significant sexual size dimorphism in *R. kukunoris* [19]; we therefore applied linear mixed models (LMMs) to examine sexual dimorphism in three morphological measurements among populations. We incorporated age as a covariate to correct its effect on body development, including SVL, LAHL, and HILL separately as dependent variables, sex as a fixed factor, and population as a random effect. The following analyses were conducted separately on males and females as the results from the LMMs showed significant morphological differences between sexes (Table 1). We then performed a partial correlation with single morphometrics and E/BS, with age as a controlled factor, in order to ensure the existence of ecogeographical adaptation in *R. kukunoris* along elevation.

To examine how geographic variation in morphology could be shaped by environmental and biological factors, we conducted hierarchical partitioning [53] using the ‘*hier.part*’ R package [54]. This method is capable of identifying the importance, effect, and independent contribution of each predictor on each morphological trait, from a combined effect of evapotranspiration, precipitation, temperature, primary productivity, and age. Hierarchical partitioning for the morphometrics and other variables was done using linear regression model of all variables. Multiple regression was as well conducted to determine morphological variation explained by the combined effects, using ANOVA for the significance testing. Owing to the high correlation between temperature and PET, we performed hierarchical partitioning with temperature only as it was suggested to be a better fit to the data than PET was (lower *p*-value on the F-test of regression lines). Following MacNally [55], a randomization test included in the R package evaluated the statistical significance for each independent contribution and yielded Z-scores for it based on an upper 0.95 confidence limit. All statistical analyses were performed in R (v. 4.0.3) and visualized using Origin 2017 software. All probabilities were two-tailed.

## 3. Results

The LMMs showed the existence of sexual dimorphism in all examined morphological traits of *R. kukurnoris*. All morphometrics significantly differed among populations and between sexes (Table 1). The results remained robust after controlling the significant effect of age (*p* < 0.01). All environmental variables displayed a negative clinal pattern against the elevational gradient (Figure 2; Appendix A). Male SVL increased along elevation (Figure 3a), while no significant changes were found in male extremities (Figure 3b,c). In females, significant elevational positive trends were observed for the two extremities (Figure 3f,g), but no obvious geographic variation was detected in SVL (Figure 3e). Although neither males nor females concurrently exhibited clinal variation in SVL and extremities, E/BS decreased along elevation in both sexes (Figure 3d,h).

Multiple regression models showed a significant correlation between the mixed effects and all morphometrics in both sexes (Table 2). Hierarchical partitioning revealed that age, NDVI, and precipitation exhibited positive impacts on growth in examined traits, while temperature showed a negative effect (Figure 4). Both sexes showed similar correlation slopes (effect) of each environmental variable, but the independent contribution differed among them. All factors (age, NDVI, precipitation, and temperature) were shown to be significant contributors in males. In females, the key factors for SVL were age and NDVI, and those for extremities were NDVI and precipitation (Table 2). Temperature was the greatest contributor in all male traits (63.37%, 47.80%, and 43.57% on SVL, LAHL, and HILL, respectively; Figure 4), whereas age and precipitation were the most influential on female SVL and extremities, respectively (32.24% on SVL and 53.93% and 48.66% on LAHL and HILL, respectively; Figure 4).

## 4. Discussion

Species’ adaptation is important for their long-term persistence. It relates to the degree to which an organism is likely to suffer harm owing to perturbation or stressor exposure [56,57]. Our results demonstrated that the morphological development of the plateau brown frog was strongly correlated to primary productivity, temperature, and precipitation, which varied along an elevational gradient on the Qinghai–Tibetan Plateau. After considering the significant effect of sexual dimorphism and age, we found a significant decrease in E/BS along elevation in both sexes. Therein, only male body size and female extremities displayed significant variations, hinting that they may adopt different strategies to achieve heat preservation. Furthermore, the decreasing E/BS along elevation, with the strong association between morphometrics and climatic factors in both sexes, provided evidence for ecogeographical adaptation in *R. kukunoris*. Overall, these results validate our predictions that both environmental and biological factors contribute to the morphological variation of *R. kukunoris* along an elevational gradient (Hypothesis 2), and ecogeographical adaptation appears in *R. kukunoris* as a joint change between body size and extremities (Hypothesis 1).

Different types of Bergmann’s cline and Allen’s cline have been reported in various intraspecific studies of anurans. For example, ecogeographical adaptation only appeared in a single sex (e.g., [58]), or the species followed an inverse pattern (e.g., [21]). The obscured presence of the ecogeographical adaptation in anurans might be attributed to their reproductive characteristics of both body size [40,59] and extremities [60], which were rare in the original study target of Bergmann’s rule and Allen’s rule. This might increase the complexity in observing the ecogeographical pattern in anurans under a potential trade-offs and stress from both reproductive investment and environmental adaptation, since organisms cannot devote extra resources to a characteristic without diverting a proportional amount of energy away from another attribute [40]. Due to environmental and reproductive factors, we suspect that thermoregulatory adaptation might appear in an interactive change between body size and extremities under relative selection pressures on each morphometric. Our study first showed that *R. kukunoris* displayed sexual dimorphism, in which body size, forelimb, and hindlimb were significantly different between sexes (Table 1). This represents a sexual difference in growth rate and age at maturity [34,35]. Then, a significant decrease in E/BS along the elevational gradient in both sexes was found after controlling the age effect (Figure 3d,h), in which significant clinal variation was observed in female extremities (Figure 3f,g) and male body size (Figure 3a). The decrease in E/BS represented a significant change between an increasing body size against extremity length or a decreasing extremity length against body size. Even though the Bergmann’s cline and Allen’s cline were not observed together in both sexes, these changes would still be advantageous owing to the corresponding low surface to volume ratios for body temperature maintenance according to the theoretical basis of the two rules [17,18].

However, the respective modification of body size and extremities between males (Figure 3a–c) and females (Figure 3e–g) might suggest that they adopt different strategies to allocate resources for the growth of different body parts and to achieve thermoregulatory and reproductive advantages. Body size and extremities have been found to be associated with reproductive strategies in both sexes that involve territoriality, fecundity, and mating success [58,61]. For male anurans, research has shown important impacts in female sexual selection and mating success from body size and extremities [39,61]. The increasing body size along elevation (Figure 3a), may help male *R. kukunoris* to gain benefits from reproduction [61] and thermoregulation [17] at the same time. On the other hand, although male extremities have been linked to reproductive importance, most of the studies display mating benefits from limb muscle mass instead of limb length [59]. Further, male-biased sexual dimorphism in limb length was rarely reported in anurans, except for the locomotor performance studies in the genus *Xenopus* (e.g., [62,63]), a genus of fully aquatic frogs. In this case, the extremity length in male *R. kukunoris* should decrease with increasing elevation and increasing muscle mass for improved reproductive and thermoregulatory performance. Although we may not explain for our results that the male extremity length remained stable along elevation (Figure 3b,c), the significant decrease in E/BS suggests a difference in resource allocation for male body size and extremities that could help to achieve better reproductive and thermoregulatory performance. With the significant association between the three morphometrics and environmental variables revealed by the hierarchical partitioning analysis (Table 2), especially the dominant effect of temperature, ecogeographical adaptation and our Hypothesis 1 can therefore be further consolidated.

Furthermore, female extremities decreased significantly along the elevation gradient, while female body size did not show clear clinal variation (Figure 3e–g). In female anurans, body size has been found to have a close relationship with maternal investment [42,58]. Previous research has also revealed that female size has a strong contribution to reproductive traits, in which egg size, clutch size, and total reproductive investments increase with female size [64]. Differently, biological importance was rarely reported in female extremities. Therefore, it is more likely to have less resource allocation for the growth of extremities compared to body size. With the significant relationship between SVL and primary productivity (Table 2), the unresponsive body size along elevation may be restricted by the lower food availability at a higher elevation [28,30]. Even though we suggest that more resources were allocated to body growth than to extremities, female frogs are found to have a higher resource requirement in body maintenance [65]. Moreover, there was sampling bias towards males in this study, and the female sample size was small at a lower elevation (Appendix A). These may explain the absence of increasing body size against elevation in female *R. kukunoris*. The hierarchical partitioning analysis revealed that primary productivity was the only significant environmental contributor in female body size, while precipitation was the dominant contributor in female extremities. Since precipitation could help to reduce the air temperature, it could also delay sexual maturity, which alleviates the fostered growth rate by increasing temperature [66]. In females, only extremities display an association with thermally related environmental variables. The environmental relatedness in extremities echoes the common garden results by Alho et al. [25], finding that the genetic cline in extremities followed an Allenian pattern under some environmental conditions, though the primary purpose of legs should not be thermoregulation. Similar to males, the significant change in E/BS shows a difference in resource allocation and provides evidence that extremities and body size collaborate to achieve ecogeographical adaptation under stress from thermoregulation and reproduction investment.

Being an indeterminate grower [27], anurans were found to be following the von Bertalanffy growth model [67], which is composed of age, growth rate, and initial body size. In other words, the increase in longevity could allow individuals to grow accordingly, which would make age one of the significant biological contributors to anurans’ growth. Thus, this could explain the strong effect of age on all morphometrics (Table 1), and on most morphometrics when they involved a combined effect (Table 2). The strong and positive association between morphological development and NDVI (Table 2) was consistent with the primary productivity hypothesis [28], which implies that body size must be sustained by an adequate food supply and predicts larger body sizes in more productive areas [28,30]. With reference to the restraining effect of temperature detected by hierarchical partitioning (Table 2), it can be explained by the temperature–size rule that refers to a decrease in adult body size by early maturity through increasing growth rate and development under high temperatures [65]. For precipitation, our results showed a positive link with growth that is contrary to the prediction from the water availability hypothesis [9]. In more detail, the positive influence of precipitation brought by the rainwater might exert a catalytic force on vegetation growth (primary productivity) and give rise to the biomass of the habitat [68,69]. Thermally, it could also alleviate desiccation and temperature, which leads to a beneficial delay in growth rate as discussed [65]. Moreover, the importance and contribution of the biological and environmental variables vary greatly between male and female *R. kukurnoris* (Table 2). Interestingly, temperature was found to be the strongest contributor in every male morphometric, while it did not have a significant impact on any female morphological trait. Herein, the most powerful explanatory variables for female body size and extremities were age and precipitation, respectively (Table 2). This situation may be explained by the sex difference in habitat use [70,71,72], which suggests that males or females may have a higher preference or need in terms of food requirements for reproductive investment, desiccation risk, or territoriality defense [72,73].

## 5. Conclusions

Ecogeographical adaptation in anurans does not follow a general pattern in distinct species [8,9,10,19,20,21]. The implication of our study is to provide insights into the enigmatic mechanism behind ecogeographical adaptation in anurans. With reference to our results, the significant association between the environmental factors and morphometrics, together with the decreasing E/BS along elevation, our Hypothesis 1 that ecogeographical adaptation may occur in an interactive change between body size and extremities is supported. The significant effects of sex and age and the results from hierarchical partitioning relate to Hypothesis 2, i.e., geographic variation in anurans’ morphology is an outcome of combined environmental and biological effects. Although age and sex were the only biological factors included in our study, we discussed several reproduction-related explanations for the sexual differences in the morphological variation of *R. kukunoris*. However, the explanatory power of the biological factors and our statistical models could be improved by adding or substituting with other biological variables that are also proven to remarkably affect anurans’ morphological growth, such as the length of the juvenile period or first-year growth [74] and age at sexual maturity [75]. The validity of our explanation may require future studies to include a reproduction-related factor for further examination. In conclusion, our results emphasize that geographic variation in anurans’ morphology should be viewed as a consequence or trade-off between the environmental effect and biological issues. The ratio between extremities and body size could be taken into account for the future study of ecogeographical patterns in anurans. However, we suggest future investigations to involve more species and with a greater geographical gradient, which would help to enhance the validity of using E/BS in ecogeographical studies of anurans.

## Figures and Tables

**Figure 1 biology-10-01081-f001:**
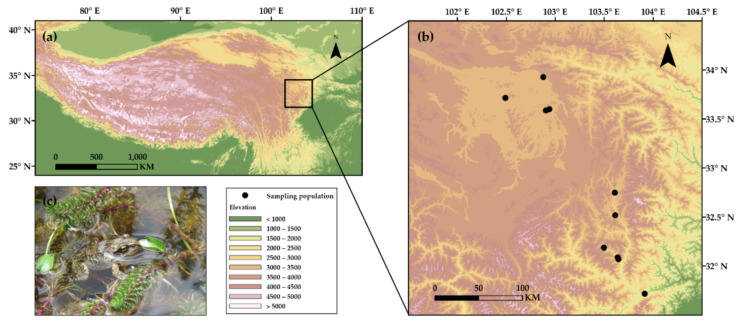
(**a**) Geographical location of the study area on the Qinghai–Tibetan Plateau, (**b**) the 10 sampling populations of the plateau brown frog, *Rana kukunoris*, and (**c**) photo of a plateau brown frog (by J. Hu).

**Figure 2 biology-10-01081-f002:**
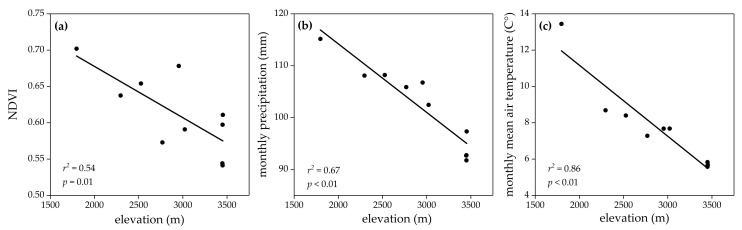
Variation in environmental variables along an elevation gradient on the eastern margin of the Qinghai–Tibetan Plateau. (**a**) Normalized difference vegetation index (NDVI); (**b**) monthly precipitation; (**c**) monthly mean air temperature.

**Figure 3 biology-10-01081-f003:**
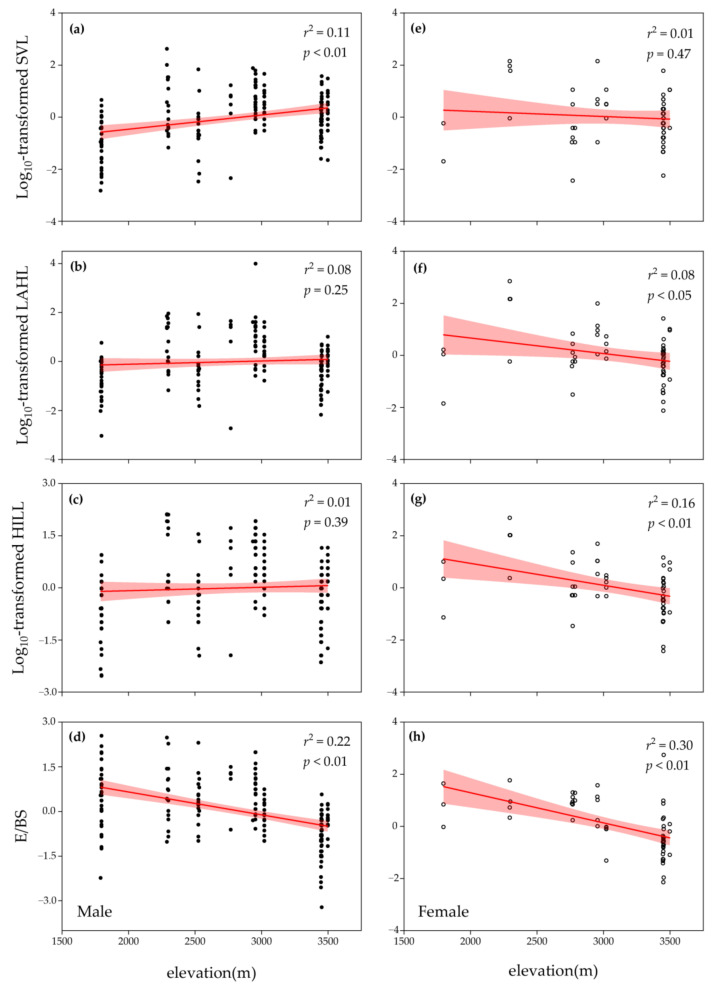
Variation in *Rana kukunoris* morphometrics along an elevational gradient on the eastern margin of the Qinghai–Tibetan Plateau. Linear regressions for each morphometric and age were used to control the effect of age in both sexes. The standardized residuals from the regression were used in another linear regression against elevation. Results are presented in *r*^2^ and *p*, respectively. Filled circles represent male data and empty circles represent female data. (**a**,**e**) Log_10_-transformed SVL; (**b**,**f**) log_10_-transformed LAHL; (**c**,**g**) log_10_-transformed HILL; (**d**,**h**) extremities to body size ratio.

**Figure 4 biology-10-01081-f004:**
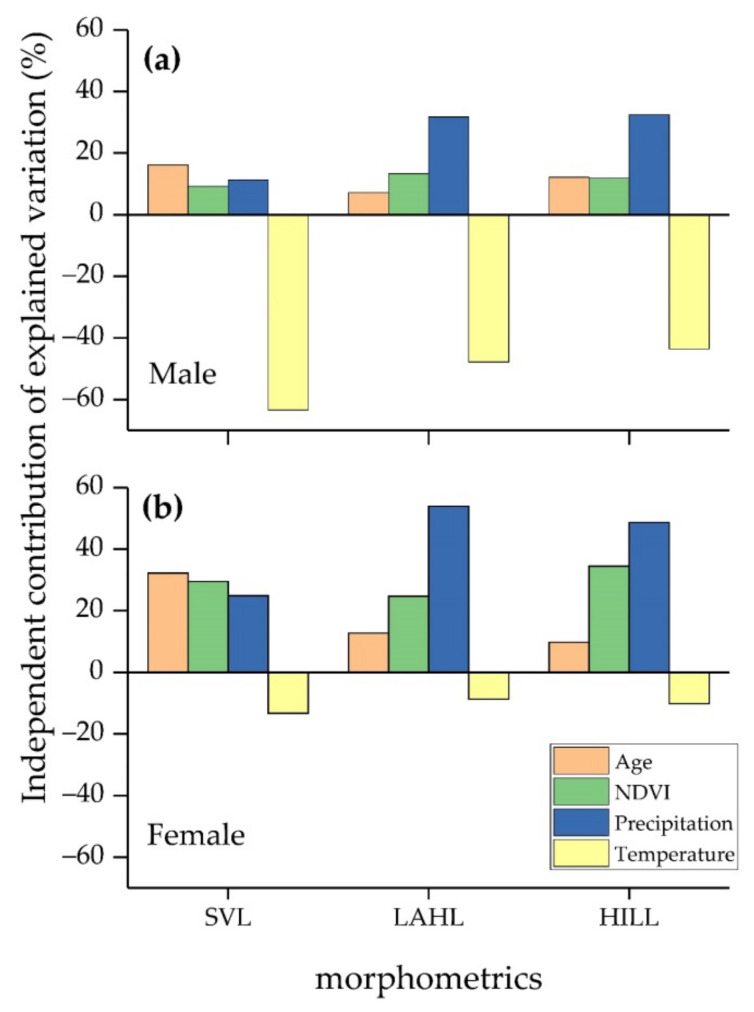
Independent contribution for each biological and environmental effect on *Rana kukunoris* morphometrics. (**a**) Effects on male morphometrics; (**b**) effects on female morphometrics.

**Table 1 biology-10-01081-t001:** The effects of sex and age on *Rana kukunoris* morphometrics. Statistics were calculated with linear mixed model, incorporating morphometrics as dependent variable, sex as fixed factor, population as random effect, and age was added as covariate. Results are shown as standard error (SE), z score (Z), and f value (F).

Morphometric	Factor	Random Effect		Fixed Effect	
		SE	Z	SE	F
SVL	Population	<0.001	**1.964** *		
	Sex			0.006	**39.918** **
	Age			0.003	**18.113** **
LAHL	Population	0.001	**1.997** *		
	Sex			0.006	**9.089** **
	Age			0.003	**7.438** **
HILL	Population	0.001	**1.980** *		
	Sex			0.007	**8.255** **
	Age			0.003	**10.346** **

Significant results in bold, and levels of significance are shown as: *, *p* < 0.05; **, *p* < 0.01. SVL, snout–vent length; LAHL, lower arm and hand length; HILL, hindlimb length.

**Table 2 biology-10-01081-t002:** Summary of multiple regression and hierarchical partitioning results for biological and environmental effects on *Rana kukunoris* morphometrics. Statistics were calculated with multiple regression models for age, normalized difference vegetation index (NDVI), precipitation (PREP), and temperature (TEMP). For multiple regression models, explained variance is given as the adjusted R square value (*r*^2^). For the variables within the full regression models, explained variance is given as a percentage of independence (%I) based on hierarchical partitioning, with +/− indicating the direction of the relationship.

Sex (*n*)	Morphometric	Full Model (*r*^2^)	Full Model Variables (Slope, %I)			
			Age	NDVI	PREP	TEMP
Male (197)	SVL	**0.274** **	**+16.14** *	**+9.21** *	**+11.28** *	**−63.37** *
	LAHL	**0.208** **	**+7.16** *	**+13.32** *	**+31.73** *	**−47.80** *
	HILL	**0.189** **	**+12.16** *	**+11.85** *	**+32.42** *	**−43.57** *
Female (58)	SVL	**0.223** **	**+32.24** *	**+29.51** *	+24.92	−13.32
	LAHL	**0.315** **	+12.72	**+24.68** *	**+53.93** *	−8.67
	HILL	**0.345** **	+9.77	**+34.42** *	**+48.66** *	−10.16

Significant results in bold, and levels of significance are shown as: *, *p* < 0.05; **, *p* < 0.01.

## Data Availability

The data presented in this study are available in the Appendix A annexed to this article. Other data presented in this study are available on request from the corresponding author.

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
