# Peer review of "Ecogeographical Adaptation Revisited: Morphological Variations in the Plateau Brown Frog along an Elevation Gradient on the Qinghai–Tibetan Plateau"

_biology, 2021, doi:10.3390/biology10111081_

Round 1

Reviewer 1 Report

General comments

The manuscript describes an extensive investigation about ecogeographical adaptation revisited: morphological variations in the Plateau Brown Frog along an elevation gradient on the Qinghai‐Tibetan Plateau. The topic of this paper is within the scope of the journal and represents a main research field because the authors are finding new findings in this field of knowledge.

The paper is generally well structured

Below my indications:

INTRODUCTION

It is well written, it describes relevant aspects about the topic to analyze with relevant references  

MATERIAL AND METHODS

This section is well written. It is described the appropriate methodology. Figures in section are acceptable.

RESULTS

Acceptable description of the results on all the variables analyzed.

Line 244 Change in Figure 2. Elavation  by elevation; Prepcipitation by precipitation

DISCUSSION

It is well argued about hypotheses tested. It is included updated references to argue the biological aspects found in the research that allow assuming the explanation of environmental effects in the changes in Plateau Brown Frog

CONCLUSION

It is clear and suggests the development of new research

Author Response

Response to Reviewer 1 Comments

  1. General comments: The manuscript describes an extensive investigation about ecogeographical adaptation revisited: morphological variations in the Plateau Brown Frog along an elevation gradient on the Qinghai‐Tibetan Plateau. The topic of this paper is within the scope of the journal and represents a main research field because the authors are finding new findings in this field of knowledge. The paper is generally well structured

Response 1: Many thanks for the positive assessment of our manuscript.

  1. INTRODUCTION: It is well written, it describes relevant aspects about the topic to analyze with relevant references

Response 2: Thanks a lot.

  1. MATERIAL AND METHODS: This section is well written. It is described the appropriate methodology. Figures in section are acceptable.

Response 3: Thanks for the positive assessment on this section.

  1. RESULTS: Acceptable description of the results on all the variables analyzed.

Response 4: Thanks.

  1. Line 244 Change in Figure 2. Elevation by elevation; Precipitation by precipitation

Response 5: Thanks for the suggestion. It is adjusted in Figure 2, 3, 4 and S1 in the revision. [Figure 2, 3, 4 and S1]

  1. DISCUSSION: It is well argued about hypotheses tested. It is included updated references to argue the biological aspects found in the research that allow assuming the explanation of environmental effects in the changes in Plateau Brown Frog

Response 6: Many thanks for the positive assessment on the section of discussion.

  1. CONCLUSION: It is clear and suggests the development of new research

Response 7: Thanks for this positive assessment.

Reviewer 2 Report

INTRODUCTION

About the Bergmannś and Allenś rules, a more complex interpretation could be explained in terms of intraspecific variation trends between local factors and how falsifying hypothesized mechanisms. See https://doi.org/10.1111/j.1600-0706.2009.17959.x

L60 surface are to volume ratio >>> surface area to volume ratio ???

Only one sampling by locality was done??? 255 individuals are enough to prove an ecogeographical pattern??? what is the population estimation?? More biological relevance about R. kukunoris should be provided.

M&M

L142-147 Biological features of this frog could be emphasized in the introduction section to context the pertinence to evaluate the ecogeographical pattern in this species

L156 The measurements are missing. The table only shows the organism count, but I’d expect the measurements to be shown in supplementary material, at least the mean and the standard deviation for each sex in each altitude.

Fig. 3. sexes could be differentiated by color or symbols

The data dispersion is excessive, the linear regression should be shown with confidence limits ... Authors could change a boxplot (showing dots by individuals) by sex and population to exhibit how individuals and populations were sampled by sex and altitude. 

There must be a diagram/figure that shows exactly the representation for each measure.

Multiple regression is a good approach but lacks a more explanation about how sexes are differentially affected in terms of physiology, habits or behavior, that should be provided. In this sense, anything about sample size is mention, and less than 10 female frogs by population were measured.

Although residuals are the input to multiple regression and hierarchical partitioning are not show to see the distribution.

Biases of populations by each elevation are evident and lacks mention.

Clustering evaluation could be a good approach given the bias of sampling.

Author Response

Response to Reviewer 2 Comments

  1. About the Bergmannś and Allenś rules, a more complex interpretation could be explained in terms of intraspecific variation trends between local factors and how falsifying hypothesized mechanisms. See https://doi.org/10.1111/j.1600-0706.2009.17959.x

Response 1: Thank you very much for this suggestion. We agree with the paper that patterns should be labelled as trends, not rules, and studies should be focusing on falsifying mechanisms rather than simply describing patterns. This is why we focused on ecogeographical adaptation and used the term throughout the article instead of using Bergmann’s rule or Allen’s rule. The ultimate goal of our study (the two hypotheses) is concord with the conclusion in the suggested paper that ‘it is time research focuses on the mechanisms underlying Bergmann’s rule rather than constant evidence that the pattern does or does not exist.’ In this revision, we revised Introduction with reference of the suggested paper. [Lines 146]

  1. L60 surface are to volume ratio >>> surface area to volume ratio ???

Response 2: Thanks. Done. [Line 64]

  1. Only one sampling by locality was done??? 255 individuals are enough to prove an ecogeographical pattern??? what is the population estimation??

Response 3: Many thanks for this comment and the important point about sample size raised here. Yes, only one sampling was done per locality. However, in the case of this study, it seems that 255 individuals are a reasonable sample size for 10 sampling localities, with the mean value of ~25 individuals per locality. Also, sample size alone may not be the most important element for proving an ecogeographical pattern, while an extensive geographical gradient is a key to clearly show the potential variation of morphological traits. Figure 2 has shown great variations in environmental condition along the elevation on the eastern margin of the Qinghai-Tibetan Plateau. This shows that the studied geographical scale is large enough to study ecogeographical pattern for the species. Moreover, extensive sampling may exert excessive pressure on the plateau brown frog populations. For enhancing the persuasiveness, we provided suggestions in the section of conclusion that future studies should include more species and greater geographical gradient to better understand the ecogeographical pattern in anurans. [Figure 2; Lines 451-453]. If you have any additional queries, please do not hesitate to contact us.

  1. More biological relevance about R. kukunoris should be provided.

Response 4: Thanks. We have added more biological relevance about R. kukunoris with examples. [Lines 135-136]

  1. L142-147 Biological features of this frog could be emphasized in the introduction section to context the pertinence to evaluate the ecogeographical pattern in this species

Response 5: Thanks for the suggestion. We have added the pertinence biological features to help understanding the ecogeographical pattern of this species. [Lines 129-148]

  1. L156 The measurements are missing. The table only shows the organism count, but I’d expect the measurements to be shown in supplementary material, at least the mean and the standard deviation for each sex in each altitude.

Response 6: Thank you for pinpointing this. We have added the relevant information in the Supplementary Files. [Table S1]

  1. 3. sexes could be differentiated by color or symbols

Response 7: Thanks for the suggestion. The updates have been made in Figure 3, in which, the data of each sex were differentiated by symbols now. [Figure 3]

  1. The data dispersion is excessive, the linear regression should be shown with confidence limits ... Authors could change a boxplot (showing dots by individuals) by sex and population to exhibit how individuals and populations were sampled by sex and altitude.

Response 8: Thanks for the suggestions. 95% confidence intervals have been added to the linear regressions for each graph in Figure 3. [Figure 3]

  1. There must be a diagram/figure that shows exactly the representation for each measure.

Response 9: Thanks. We have newly provided a table in Supplementary Files for a detailed description of the measurements performed (Table S2). A diagram/ figure can be found in Fei et al. 2009 that we cited as reference for the definition of the measurement used. [Line 188]

  1. Multiple regression is a good approach but lacks a more explanation about how sexes are differentially affected in terms of physiology, habits or behavior, that should be provided. In this sense, anything about sample size is mention, and less than 10 female frogs by population were measured.

Response 10: Thanks for the comment. Apart from using multiple regression, we have enhanced the completeness of our explanation by adding elaboration and information in terms of physiology, habits or behavior in Discussion. [Lines 430-433]

  1. Although residuals are the input to multiple regression and hierarchical partitioning are not show to see the distribution.

Response 11: Thanks for the comments. Although hierarchical partitioning is not show to see the distribution, there may not be any statistical problem or error if the number of input variables does not exceed nine (Walsh & Mac Nally, 2020). (https://cran.r-project.org/web/packages/hier.part/hier.part.pdf)

  1. Biases of populations by each elevation are evident and lacks mention.

Response 12: Thanks for the comment. Revisions have been made in Discussion. [Lines 391-393]

  1. Clustering evaluation could be a good approach given the bias of sampling.

Response 13: Thanks for the suggestion. We agree that clustering evaluation could be a good approach for sampling bias, but our methods were better to explain for proposed hypotheses. However, the sampling bias problem is mentioned in Discussion. [Line 391-393]

Reviewer 3 Report

I refrain from specific comments (apart from the language editing done in the pdf document) until the issues of ethical concern (why are 3 out of 4 authors of the paper reporting the original data Feng et al 2015 not considered as coauthors in this paper, they collected the data) and of data treatment ("curation") (figures 3ab in this paper should be identical with fig 2b in Feng et al. 2015, but they are not) are resolved satisfactorially.

Author Response

Response to Reviewer 3 Comments

  1. I refrain from specific comments (apart from the language editing done in the pdf document) until the issues of ethical concern (why are 3 out of 4 authors of the paper reporting the original data Feng et al 2015 not considered as coauthors in this paper, they collected the data) and of data treatment ("curation") (figures 3ab in this paper should be identical with fig 2b in Feng et al. 2015, but they are not) are resolved satisfactorily.

Response 1: a) Thanks a lot for this reminding, and sorry for the deficient expression in the previous manuscript. In fact, there should not be any ethical issue. Mr. Feng joined Hu’s lab in Chengdu Institute of Biology, Chinese Academy of Sciences (CAS), in 2013 and finished his master’s dissertation in 2015, supervised by Prof. Hu. During these years, Feng took part in the project of the plateau brown frog (Rana kukunoris) with the funding from Hu’s lab. Therefore, Hu’s lab owned the original data of the plateau brown frog, but not Feng. Moreover, according to the policy of the CAS, the copyright of any data and result of graduate’s dissertation belong to the CAS and the original research team/laboratory. Undoubtedly, Prof. Junhua Hu as the head of the research team, he has the right to use and publish these data when some new members join his lab. If you have any additional queries, please do not hesitate to contact us.

  1. b) Thanks for pinpointing this. We did a check on the data, statistics and graphs. Although Figures 3 a & e in this study are not identical with Figure 2b in the previous paper, they are similar with each other. For the slight differences, they were resulted from the different methods in respective linear regression. Age effect was controlled in the data presented in Figures 3 a & e of this study, while figure 2 B in Feng et al. 2015 used temperature, altitude and age as covariates.

  1. L15 ‘gain’ changes to ‘improve’; delete ‘advantage’

Response 2: Revised. [Line 16]

  1. L16 ‘possessed’ changes to ‘showed’

Response 3: Revised. [Line 17]

  1. L18 ‘dropping’ changes to ‘decreasing’

Response 4: Revised. [Line 19]

  1. L22 ‘product’ changes to ‘result’

Response 5: Revised. [Line 22]

  1. L24 ‘Anurans’ changes to ‘Several anurans’

Response 6: Revised. [Line 26]

  1. L27 ‘process’ changes to ‘mechanisms’

Response 7: Revised. [Line 29]

  1. L32 ‘the validation of’ changes to ‘potential’

Response 8: Revised. [Line 35]

  1. L34 ‘possessed’ changes to ‘showed’

Response 9: Revised. [Line 34]

  1. L35 ‘while’ changes to ‘whereas’

Response 10: Revised. [Line 38]

  1. L50 ‘scaleless and highly permeable nature’ changes to ‘water permeable skin’

Response 11: Revised. [Line 53]

  1. L52 ‘them advanced’ changes to ‘increased’

Response 12: Revised. [Line 55]

  1. L59 add ‘.’ After ‘[e.g., 13-16]’

Response 13: Revised. [Line 62]

  1. L76 delete ‘anurans’’; add ‘of anurans’ after ‘morphological development’

Response 14: Revised. [Line 79]

  1. L79 ‘loosen’ changes to ‘untighten’

Response 15: Revised. [Line 83]

  1. L81 delete ‘are more favourable to’; ‘surviving’ changes to ‘survive better’

Response 16: Revised. [Line 85]

  1. L95 ‘corresponding’ changes to ‘correspondence’

Response 17: Revised. [Line 98]

  1. L98 ‘context’ changes to ‘constraints’

Response 18: Revised. [Line 101]

  1. L101 Please explain clearly on “reproductive functions”

Response 19: Thanks. We have replaced it with ‘reproductive benefits’ that would be clearer and close to what we mean. [Line 105]

  1. L107 add ‘,’ after ‘However’

Response 20: Revised. [Line 107]

  1. L108 Please replace with a better and more elusive terms for “reproductive importance”

Response 21: Revised. [Line 111]

  1. L111 ‘collaborative’ changes to ‘joined’

Response 22: Revised. [Line 115]

  1. L113 Please replace with a better and more elusive terms for “reproductive importance”

Response 23: Thanks, it is replaced with ‘reproductive benefits’. [Line 117]

  1. L124 ‘fit’ changes to ‘model organism’

Response 24: Revised. [Line 143]

  1. L125 ‘obscured’ changes to ‘enigmatic’

Response 25: Revised. [Line 148]

  1. L127 ‘appear’ changes to ‘is evident’

Response 26: Revised. [Line 149]

  1. L146 add ‘,’ after ‘thus’

Response 27: Revised. [Line 140]

  1. L147 delete ‘to enter’

Response 28: Revised. [Line 140]

  1. L156-159 “you are using and analysing the same data presented in Feng et al 2015!?

Response 29: Thanks for the question. We only used age and SVL which were the same as Feng et al 2015, and other data used in this study are new, unpublished and collected by Hu’s lab.

  1. It seems strange to me that besides Junhua HU the authors of the original study are not considered in this study although they did the field work and skeletochronological age determinations.”

Response 30: Thanks a lot for pinpointing this. We have detailly explained this issue in “Response 1”. If you have any additional queries, please do not hesitate to contact us.

  1. L162-163 “This means that you measured apart from the SVL already used for Feng et al. 2015 exclusively limb lengths.”

Response 31: Yes, we did.

  1. L168 Meaning of “diminished”

Response 32: Thanks for this comment. To be clearer, we changed ‘diminished’ to “reduced”. [Line 195]

  1. L199 Why use “linear regression” to see variation?

Response 33: Thanks for the comment. The sentence has been corrected. [Line 228]

  1. 3 Why are figures 3A and B different from figure 2 B in Feng et al. 2015, if the same material was used?

Response 34: Thanks for pinpointing this. We did a check on the data, statistics and graphs. Figures 3 a & e in this study are similar with Figure 2b in the previous paper, although they are not the same. For the slight differences, they were resulted from the different methods in respective linear regression. Age effect was controlled in the data presented in Figures 3 a & e of this study, while figure 2 B in Feng et al. 2015 used temperature, altitude and age as covariates. In this revision, we also checked and corrected the information across panels in Figure 3. [Figure 3]

  1. L470 ‘2005’ changes to ‘2015’

Response 35: Revised. [Line 512]

Round 2

Reviewer 1 Report

The document was satisfactorily enriched with the modifications made.

Author Response

Many thanks for the positive assessment of our revision.

Reviewer 2 Report

All suggestions were attended. Thus the manuscript is more appropriate to be published. However, I think that the results section can be improved with the brief mention of data measures itself in relation to elevation due all text does not provide some reference about the magnitude of size change in extremities.

Author Response

Thanks a lot for your positive assessment of our revision. And, we have updated results section with brief mention of data measures in relation to elevation. [Lines 245-250]

Reviewer 3 Report

The ethical issues raised by me are now settled, your explanations are satisfactory. Now, reading carefully introduction, discussion and conclusions I find that you did not mention several recent publications on the same issue (ecogeographical adaptations of anurans), which show exactly the joined effect of environmental factors and biological ones, like the life-history trait fecundity (see comment on the revised ms). I believe that they showed be discussed as well.

Author Response

  1. The ethical issues raised by me are now settled, your explanations are satisfactory. Now, reading carefully introduction, discussion and conclusions I find that you did not mention several recent publications on the same issue (ecogeographical adaptations of anurans), which show exactly the joined effect of environmental factors and biological ones, like the life-history trait fecundity (see comment on the revised ms). I believe that they showed be discussed as well.

Response 1: Thanks a lot for the reminding. We have accordingly updated Introduction and Conclusions. [Lines 85-90, 425-426]

  1. L376 This is only true in part. Most of SVL, sometime nearly complete adult size is achieved during the period between metamorphosis and onset of sexual maturity. Later growth is negligible in most anuran species (there is a lot of literature on this). So it is not really age, but the length of the juvenile period.

Response 2: Thanks for pinpointing this. We have accordingly revised the contents in Discussion and Conclusion. [Lines 384-385, 420-424]

  1. L383 this is the issue put forward above.

Response 3: Thanks. Revised. [Lines 384-385, 420-424]

  1. L400 “has been disputable in the past two decades” changes to “does not follow a general pattern in distinct species”

Response 4: Thanks. Done. [Lines 409-410]

  1. L412-414 Not really new. See papers on natterjack toads for latitudinal and altitudinal size variation: Oromi, N.; Sanuy, D.; Sinsch, U. Altitudinal variation of demographic life-history traits does not mimic latitudinal variation in natterjack toads (Bufo calamita). Zoology 2012, 115, 30-37, doi:10.1016/j.zool.2011.08.003. and find the related article like this paper to enrich your paper.

Response 5: Thanks for the comments. Referring to these, revisions were made in Introduction and Conclusion to enrich our paper. [Lines 85-90, 425-426] See also Response 1.
